# Orchestrating Resilience: How Neuropilin-2 and Macrophages Contribute to Cardiothoracic Disease

**DOI:** 10.3390/jcm13051446

**Published:** 2024-03-01

**Authors:** Rajeev Dhupar, Amy A. Powers, Seth H. Eisenberg, Robert M. Gemmill, Charles E. Bardawil, Hannah M. Udoh, Andrea Cubitt, Leslie A. Nangle, Adam C. Soloff

**Affiliations:** 1Department of Cardiothoracic Surgery, University of Pittsburgh School of Medicine, Pittsburgh, PA 15213, USA; dhuparr2@upmc.edu (R.D.); hmu5@pitt.edu (H.M.U.); 2UPMC Hillman Cancer Center, University of Pittsburgh, Pittsburgh, PA 15213, USA; 3Surgical and Research Services, VA Pittsburgh Healthcare System, Pittsburgh, PA 15240, USA; 4Division of Hematology/Oncology, Department of Medicine, Medical University of South Carolina, Charleston, SC 29425, USA; gemmill@musc.edu; 5Hollings Cancer Center, Medical University of South Carolina, Charleston, SC 29425, USA; 6aTyr Pharma, San Diego, CA 92121, USA; acubitt@atyrpharma.com (A.C.); lnangle@atyrpharma.com (L.A.N.)

**Keywords:** neuropilin-2, macrophages, semaphorins, VEGF, immunity, non-tyrosine kinase receptors, lung cancer, cardiac disease, asthma, cardiothoracic disease

## Abstract

Immunity has evolved to balance the destructive nature of inflammation with wound healing to overcome trauma, infection, environmental insults, and rogue malignant cells. The inflammatory response is marked by overlapping phases of initiation, resolution, and post-resolution remodeling. However, the disruption of these events can lead to prolonged tissue damage and organ dysfunction, resulting long-term disease states. Macrophages are the archetypic phagocytes present within all tissues and are important contributors to these processes. Pleiotropic and highly plastic in their responses, macrophages support tissue homeostasis, repair, and regeneration, all while balancing immunologic self-tolerance with the clearance of noxious stimuli, pathogens, and malignant threats. Neuropilin-2 (Nrp2), a promiscuous co-receptor for growth factors, semaphorins, and integrins, has increasingly been recognized for its unique role in tissue homeostasis and immune regulation. Notably, recent studies have begun to elucidate the role of Nrp2 in both non-hematopoietic cells and macrophages with cardiothoracic disease. Herein, we describe the unique role of Nrp2 in diseases of the heart and lung, with an emphasis on Nrp2 in macrophages, and explore the potential to target Nrp2 as a therapeutic intervention.

## 1. An Introduction to Neuropilins

Neuropilins are non-tyrosine kinase receptors belonging to a family of type I transmembrane glycoproteins with a broad tissue distribution in vertebrates [1,2]. In humans, the neuropilin family consists of two members: neuropilin-1 (Nrp1) and neuropilin-2 (Nrp2) that share a 44% amino acid sequence homology [3]. Nrp1 was identified in 1991 as a receptor for class 3 semaphorins (Sema3) mediating chemorepulsive guidance of developing axons, with its homolog Nrp2 described six years later [4,5,6]. Neuropilins are expressed in endothelial, epithelial, and immune cells, and regulate angiogenesis, lymphangiogenesis, and immunity via cell survival, migration, vesicular trafficking, and proliferation. As such, neuropilins are integrally involved in organ development, homeostasis, and the development of diseases ranging from cardiovascular issues to cancer [7,8,9]. Although Nrp1 has been studied extensively [10,11,12], much less is known about Nrp2.

Mouse studies have begun to reveal the importance of Nrp2 in development. Genetic ablation of Nrp2 reduces the formation of lymphatic capillaries and results in abnormal arrangement of cranial and spinal nerves [13,14]. Mice deficient in Nrp2, homozygous Nrp2-/- (Nrp2^KO^) lack responsiveness to repulsive guidance events mediated by Sema3F. Nrp2^KO^ mice have several major fiber tracts in the brain that are either severely disorganized or missing. Nrp2 deficiency also results in the separation of axons from a fascicle of the oculomotor nerve as well as the absence of the peripheral projection of the trochlear nerve. Conditional loss of Nrp2 impairs sensorimotor learning without affecting goal-directed instrumental learning. In contrast, depletion of Nrp1 in mice is embryonically lethal at E10.5 to E13.5 due to deficiencies in neuronal vascularization, aortic arch malformation, and disorganized yolk sac vascularization [15,16,17]. Dual knockout of both Nrp1 and Nrp2 creates a more severe phenotype, with mice dying in utero at E8.5, displaying vascular defects and areas in the yolk sac devoid of blood vessels [15].

## 2. Neuropilin Structure and Variants

Neuropilins are expressed on the cell surface, cytoplasm, mitochondria, nucleus, and may be released in soluble forms [3,18,19]. Although there is only one Nrp1 variant, there are six variants of Nrp2 (Nrp2a22, Nrp2a17, Nrp2a0, Nrp2b5, Nrp2b0, and secreted S_9_Nrp2) generated via alternative splicing and insertion of 5, 17, or 22 amino acids in the membrane-proximal ectodomain [3]. Of these, Nrp2a17, Nrpa22, Nrp2b0, and Nrp2b5, as well as soluble S_9_Nrp2 have been observed in the lung and heart [3]. Structurally, neuropilins consists of an extracellular domain comprising two cubilin homology subunits, a factor V/VIII coagulation factor homology subunit, and a MAM subunit followed by a single-pass transmembrane domain and short cytoplasmic tail of 42–46 amino acids [3]. Nrp1 and Nrp2 may homodimerize or heterodimerize through interactions of the α-helical transmembrane domain and the MAM subunit [20,21,22,23]. Although Nrp1 and Nrp2 possess conserved functions due to binding of several shared Sema3 and vascular endothelial growth factor (VEGF) proteins, variation in the sequence and structure of the C-terminal region confers receptor-specific adaptor binding affinities and subsequent signaling between the neuropilin isoforms [3,4,23,24].

## 3. Neuropilin Signaling

Neuropilins lack endogenous kinase activity due to the absence of a cytosolic protein kinase domain, and instead serve as co-receptors to propagate signaling through intermediates (Figure 1). Neuropilin holoreceptors mediate interactions between growth factors and their canonical receptors, leading to autophosphorylation, the recruitment of adaptor proteins, and the triggering of intracellular signaling cascades. Nrp1 and Nrp2 homo- and heterodimers can bind several ligands and form supramolecular holoreceptor complexes with plexins. Signaling by Nrp2a and Nrp1 is regulated by a carboxyl terminal PDZ binding motif and its association with factors such as GIPC via a C-terminal SEA sequence. By contrast, Nrp2b shares ~11% of the intracellular and transmembrane sequence with Nrp2a and possesses a 46 amino acid cytoplasmic domain which interacts with GS3Kβ to mediate independent signaling pathways [25,26,27]. Although neuropilins interact with numerous ligands, VEGFs and Sema3s are the most widely studied, with both having different binding affinities for Nrp2. Semaphorins and VEGFs have different binding domains on neuropilins and thus, do not compete.

Neuropilins can bind directly to VEGFs, serve to bridge VEGFs with VEGF receptors (VEGFRs), or mediate ligand signaling independent of VEGF receptors [28]. While VEGF-A plays a primary role in angiogenesis, VEGF-C and VEGF-D are important for lymphatic development. Although VEGF-A binds both Nrp1 and Nrp2, VEGF-C and VEGFR-3 preferentially bind to Nrp2. Interestingly, binding to soluble Nrp2 (S_9_Nrp2) effectively blocks VEGF-C/Nrp2 interaction [29,30]. There are six known classes of semaphorins, which include mammalian Sema3A, Sema3C, and Sema3F. Homodimers of Nrp2 preferentially interact with Sema3C and Sema3F, but can bind to Sema3A as well, forming signaling complexes in the plasma membrane [23,24,28,31]. Ultimately, the function of Nrp2 is highly context dependent. For example, following VEGF-A and VEGF-C ligation, Nrp2 enhances endothelial cell survival through VEGFR-2, whereas ligation of Sema3F by Nrp2 inhibits endothelial cell migration and survival [29].

Other Nrp2 ligands include HGF, PDGF, EGFR, TGFβ, PTEN, integrins, and L1-CAM [7,32,33,34,35,36]. Binding to TGFβ results in activation of the Smad2/3 pathway exerting anti-apoptotic and anti-proliferative effects, whereas binding with PTEN inhibits the effects of the Sema3F/Nrp2 signaling. Nrp2 also binds integrin cell adhesion molecules, which, like neuropilins, lack kinase activity but regulate signaling via conformation changes that alter co-receptor interactions [37,38,39]. Nrp2 facilitates adhesion to extracellular matrix (ECM) and cellular migration by inducing integrin engagement and subsequent integrin-dependent cellular processes, and depletion of Nrp2 can reduce extracellular adhesion.

## 4. Neuropilin-2 in Immunity

The role of neuropilins is highly context dependent, and in immunity, these functions can change depending upon the state of cell differentiation/maturation, the tissue microenvironment, or type of threat encountered. This is illustrated following the inhalation of lipopolysaccharide (LPS), membrane components of Gram-negative bacteria. While LPS-mediated TLR activation attenuates Nrp1 expression in macrophages, it upregulates Nrp2 [40]. The role of Nrp1 in immunity, particularly in regulatory T cells, has been extensively studied and expertly reviewed previously [10]. By contrast, comparatively little is known about how Nrp2 regulates the immune system [9,41]. In addition to macrophages, Nrp2 is expressed by dendritic cells (DCs), T and B cells, and mast cells and is crucial for the regulation of innate and adaptive immunity. Although our understanding of Nrp2 in immunity continues to grow, there remain more questions than answers.

## 5. Neuropilin-2 in Macrophages

There is increasing interest in the role of Nrp2 in macrophages, which serve to bridge innate and adaptive immunity. Macrophages are heterogeneous and highly plastic myeloid cells that undertake an array of house-keeping functions, ranging from the clearance of cellular debris to immune surveillance. Notably, macrophages coordinate the response to inflammation and wound healing [42,43]. Nrp2 is weakly expressed on macrophage precursors in the bone marrow and monocytes. Upon monocyte extravasation into tissues and differentiation to macrophages, Nrp2 is upregulated with prominent expression in tissue-resident alveolar macrophages (AMs) of the lung [44,45]. In addition, Nrp2 has been detected in interstitial macrophages and bronchial macrophages of the lung, as well as peritoneal macrophages, intravascular macrophages, microglia (brain resident tissue macrophages), macrophage-derived osteoclasts, and tumor-associated macrophages (TAMs) [45,46]. Nrp2 is expressed along the spectrum of macrophage polarization states with reports of Nrp2-regulating functions in both tolerogenic (alternatively activated, M2-like) and inflammatory (classically activated, M1-like) macrophages.

There is evidence to suggest that the main role of Nrp2 in macrophages is to resolve inflammation. Nrp2 is upregulated on macrophages in response to inflammation, often experimentally induced via LPS administration, and is dependent upon MyD88 and NF-κβ activation [18,44,47,48]. In addition, inhaled LPS induces the release of soluble Nrp2 (S_9_Nrp2) from macrophages in the airways [44]. Mechanistically, Nrp2 inhibits NF-κβ activation, inducing expression of phosphorylated IκBα and creating a negative feedback loop to suppress acute inflammation [47]. In a model of acute lung injury, myeloid-specific ablation of Nrp2 (*Nrp2*^fl/fl^*LysM*-Cre) prolonged accumulation of monocytes, macrophages, and neutrophils in bronchoalveolar lavage associated with increased protein and CCL2 levels indicative of tissue damage in the lung [44]. Depletion of Nrp2 in myeloid cells (*Nrp2*^fl/fl^*Lyz2*-Cre mice) exacerbated inflammation and disease severity in a mouse model of colitis [47]. Further, Nrp2 potentiates the suppressive activity of CD9^+^ regulatory B cells, which was attenuated by inhibition of Nrp2 [49]. Moreover, LPS-induced activation of microglial and THP-1-derived macrophages results in the secretion of S_9_Nrp2, which subsequently attenuates further LPS-induced nitric oxide, TNFα, and IL-1β production [50].

Nrp2 has been implicated in intracellular trafficking in association with VEGF-C-induced LAMP-2, a lysosomal membrane protein involved in autophagy, and WDFY-1, a factor involved in vesicular trafficking. Notably, LAMP-2 is involved in the MHC class II-mediated presentation of exogenous antigens, suggesting that the VEGF-C/Nrp2 axis may be involved in the antigen processing in macrophages [51]. WDFY-1 also promotes the TLR3/4 ligand-induced activation of transcription factors IRF3 and NF-κβ independent of MyD88 by bridging the interactions between TLR3/4 and TRIF, resulting in the production of type I IFNs and inflammatory cytokines in response to poly(I:C) and LPS [52]. Ligation of Sema3F by Nrp2 results in the reorganization of actin filaments at the plasma membrane to increase migration, implicating Nrp2 in individual macrophage functions without generalized polarization [53].

Nrp2 is one of a small number of proteins post-translationally modified in immune cells to carry polysialic acid (PSA) [50,54]. Polysialylation occurs in the linker region of Nrp2 and can contain between 8 and up to 100 α2,8-linked PSA residues [18,50]. PSA facilitates intercellular contact, promotes signaling through steric and electrostatic exclusion, and serves as a recognition pattern and immunomodulator. Polysialylation is an important prerequisite for Nrp2-specific function in macrophages and is involved in migration, phagocytosis, antigen presentation, cell–cell and cell–matrix contact. Macrophages and DCs arise from a common monocyte dendritic cell progenitor (MDP) in the bone marrow and are closely related in form and function [55]. As such, Nrp2 may exert similar effects in macrophages as have been observed in DCs. Polysialylated Nrp2 is required for CCL21-, but not CCL19-mediated trafficking of DCs to lymph nodes, allowing for antigen presentation and immune initiation [56]. Mechanistically, polysialylation on Nrp2 has been proposed to protect against interactions with other molecules until DCs traffic to lymph nodes [56,57]. Through binding the Siglec-11 receptor on macrophages, soluble low-molecular-weight PSA inhibits inflammatory cytokine production, phagocytosis, and oxidative burst in macrophage exerting anti-inflammatory effects [58]. Removing PSA decreased CCL21 activation of JNK and Akt signaling pathways, and enhanced peritoneal macrophages’ phagocytosis of Klebsiella pneumoniae, potentiating bacterial clearance [56,59]. In addition, PSA on Nrp2 is progressively lost on monocytes and monocyte-derived cells as they migrate towards regions of inflammation within the lung and peritoneum, suggesting that the release of polysialylated Nrp2 may be an early macrophage-mediated event to resolve inflammation [59].

## 6. Neuropilin-2 in Benign and Malignant Lung Processes

### 6.1. Respiratory Infections

Pulmonary macrophages constitute the first line of cellular immune defense in the lung and are continually exposed to inhaled pathogens. As defects in innate immunity are associated with pneumonia, the role of Nrp2 as an immunoregulator in the lung has recently been scrutinized [60,61,62,63]. Through amplifying microenvironmental signals, Nrp2 affects the function of pulmonary macrophages (pMacs), including alveolar, interstitial, and recruited monocyte-derived inflammatory macrophages. This is exemplified during acute bacterial pneumonia, where excessive neutrophil infiltration may lead to unintended tissue damage. Here, increased Nrp2 expression by pMacs during both bacterial and non-bacterial pneumonia occurs through a MyD88/NF-κβ-dependent signaling pathway [64]. Expression of Nrp2 on pMacs subsequently regulates crosstalk with neutrophils to enhance their phagocytosis and bacterial killing. In a murine model of *E. coli*-induced pneumonia, conditional depletion of Nrp2 in pMacs (*Nrp2*^fl/fl^*CD11c*-Cre) resulted in pathogenic neutrophil recruitment in the lung associated with severe lung injury, edema, increased pulmonary bacteremia, and reduced survival [64]. Mechanistically, the binding of pMac-expressed Nrp2 by neutrophils in vitro promoted phagocytosis, bacterial clearance, and increased *TNFα* and *CCL2* release by neutrophils, while decreasing *CXCL2* [64]. In this manner, Nrp2 coordinates an effective pulmonary immune response and subsequent resolution between pMacs and neutrophils.

In contrast to bacterial pneumonia, Nrp2 may have a different role in viral infections. Multiple viruses, such as SARS-CoV2, EBV, HTLV-1, Lujo virus, and HCMV utilize neuropilins as host receptors or co-receptors for entry [65,66,67]. This has been shown to be regulated, in part, by binding of the viral CendR motif to the b1 domain of neuropilin facilitating infectivity. Nrp2 has been identified as the receptor for the HCMV pentamer glycoprotein complex, facilitating infection of epithelial/endothelial cells [68]. In Lujo virus, the GP1 receptor binds Nrp2, mediating cellular entry. Lujo virus results in respiratory distress, neurological problems, and circulatory issues that lead to death. Although not investigated, one can speculate that upregulation of Nrp2 in response to viral-induced inflammation in endothelial cells, epithelial cells, and macrophages may further the spread of such viruses and underly their resultant pathology.

### 6.2. Asthma

Asthma is a heterogeneous disease characterized as either type-2 high with increased airway eosinophilia and Th2 cytokines, including IL-4, IL-5, and IL-13, or type-2 low, which lacks airway eosinophils and other markers of type-2 high disease [69,70,71]. Within the type-2 low asthma population, there is a subset of patients that experience neutrophilic airway inflammation that is resistant to glucocorticoid treatment, can be more severe, and presents a therapeutic challenge for clinicians [71,72,73]. Neuropilin-2 may play an important role in this disease process and even represent a therapeutic target.

Mice with experimental neutrophilic asthma also have high numbers of pMacs and DCs that express Nrp2, and increased soluble S_9_Nrp2 is found in bronchoalveolar lavage fluid [74]. In this study, the ablation of Nrp2 in myeloid cells (*Nrp2*^fl/fl^*LysM*-Cre) resulted in increased neutrophils and lymphocytes in neutrophilic, but not eosinophilic, asthma with no difference in mucus production or airway resistance [75]. Additionally, the TLR response to inhaled bacteria can potentially induce asthma exacerbations [71,74,76,77]. Because Nrp2 is a regulator of TLR-mediated inflammatory responses within the lung, it may also be an effective target for treatment of type-2 low neutrophilic asthma exacerbations [78]. Alternatively, because Nrp2 is highly expressed on smooth muscle cells (SMCs) of the colon, and conditional knockout of Nrp2 in SMCs (*Nrp2*^fl/fl^*SM22a*-CreER^T2^) leads to increased contractility [79], Nrp2 may play a role in SMC contractility during asthma attacks. Interestingly, although the chemokines CXCL1, CXCL2, and CXCL5 as well as the leukocyte adhesion ICAM-1 are important for recruitment of neutrophils to inflamed asthmatic airways, ablation of Nrp2 does not alter chemokine expression within the lungs. This suggests that Nrp2 inhibition of neutrophil recruitment occurs through alternative pathways [80,81].

As an alternate mechanism related to pneumonia, Nrp2 may play an important role in the pMac-mediated clearance of debris and dead cells of the alveoli, known as efferocytosis. Efferocytosis is necessary for the resolution of inflammation, and impairment of this process has been associated with severe and steroid refractory asthma [82,83,84,85]. Unique macrophages expressing Nrp2, and specifically the Nrp2b isoform, may be specialized for pro-efferocytic function [46,86]. In mice with neutrophilic asthma, ablation of Nrp2 in pMacs results in impaired efferocytosis [75]. Decreased clearance of apoptotic cells may contribute to increased airway inflammation due to secondary necrosis of neutrophils and the subsequent increased release of damage-associated molecular patterns (DAMPs) [87]. Although efferocytosis induces the expression of the immunosuppressive cytokines IL-10 and TGFβ, Nrp2^KO^ mice produced similar IL-10 and TGFβ levels as wild types, suggesting that the anti-inflammatory effect of Nrp2-mediated efferocytosis is distinct from these cytokines [87].

### 6.3. Lung Cancer

In lung cancer, Nrp2 expression is correlated with advanced disease stage at diagnosis, epithelial-to-mesenchymal transition (EMT), metastasis, decreased survival, and drug resistance [88,89,90,91,92]. Nrp2 is upregulated by epithelial cells upon neoplastic transformation, where it supports cancer cell survival and metastasis by promoting resistance to systemic treatments, angiogenesis, and lymphangiogenesis [93,94,95,96]. Within patients with lung cancer and mouse models, Nrp2 isoforms can have markedly different effects on tumor biology. Recently, a tumor-promoting role of Nrp2b has been described with expression associated with lung cancer development, progression [89,90], advanced stage, worse progression-free survival, increased PD-L1 levels, and acquisition of EGFR inhibitor resistance [90,97,98]. By contrast, Nrp2a expression on cancer cells inhibits these functions [90]. Knockdown of either total Nrp2 or Nrp2b in lung cancer models significantly inhibited tumor formation, whereas Nrp2a knockdown had only a modest effect [90]. Nrp2 isoforms in lung cancer differentially effect MET signaling and recruitment of GIPC1 and PTEN [90]. TGFβ is an integral driver of EMT, promoting lung cancer metastasis, and Nrp2 is correlated with EMT phenotypes [90]. Here, TGFβ upregulates Nrp2 and preferentially Nrp2b, which enhanced cellular migration, invasion into Matrigel, and tumorsphere formation in lung cancer cells, as well as promoted metastasis in lung cancer xenograft mouse models [90,91]. Conversely, overexpression of Nrp2 in epithelial cells activates TGFβ signaling, leading to the phosphorylation of the Smad2/3 complex, inhibition of E-cadherin (epithelial marker), and increased expression of vimentin (mesenchymal marker) [99].

Further details of the role of Nrp2 in cancer have been described in other malignancies [91,100,101,102,103,104]. Nrp2 enhances cancer cell survival through the regulation of autophagy and endocytic trafficking, a function distinct from its role as a co-receptor [105]. Ligation of VEGF-C by Nrp2 promotes autophagy via inhibition of mTORC1 to confer resistance to metabolic stress encountered by radiation and chemotherapy [106]. Additionally, Nrp2 was shown to mediate VEGF-C-induced activation of mTORC2 and phosphorylated Akt (Ser 473) to escape oxidative stress [107]. Interestingly, in the presence of hypoxia and nutrient deprivation, Nrp1 is degraded via autophagy, whereas Nrp2 persists and can function independently to mediate endothelial tube formation [108]. Nrp2 regulates the secretory function and exocytosis, including vesicular trafficking and endosomal recycling processes [86,105]. Secretory functions in cancer cells may enhance their survival, confer resistance to therapy, and promote cell-to-cell communication in a paracrine manner within the tumor microenvironment [109].

Interestingly, Nrp2 affects the expression of proteins important in systemic therapy. The interaction of VEGF-C with Nrp2 induces expression of immunosuppressive PD-L1, associated with immune cell exclusion and resistance to checkpoint inhibitor therapy [110]. Here, VEGF/Nrp2 signaling contributes to PD-L1 expression on cancer cells through the activation of the guanosine triphosphatase (GTPase) Rac1 and the transcriptional coactivators YAP/TAZ [110,111]. In conjunction, TAZ confers stem cell-like properties in cancer cells following VEGF/Nrp2-mediated activation of Rac1 [111]. Additionally, EGFR is a targetable mutation in some lung cancers, and treatment with EGFR-targeted antibodies confer significant improvement in survival. Nrp2 expression is inversely associated with EGFR expression in cancer cells and inhibits the rescue pathways within EGFR- or MET-addicted carcinoma cells, likely through NF-κβ repression [47,112]. Loss of the tumor suppressor PTEN, which slows cell division, repairs DNA damage, and regulates apoptosis, is associated with increased Nrp2 in high-grade tumors [113]. Nrp2 expression is also associated with decreased IGF-IR in PTEN^Null^ tumors, and Nrp2 ablation in cancer cells restored the efficacy of IGF-IR-targeted therapy [114]. Lastly, Nrp2-mediated regulation of the androgen receptor (AR), a ligand-dependent nuclear transcription factor, is implicated in prostate cancer progression and resistance to therapy. Upon binding VEGF-C, the Nrp2b isoform, but not Nrp2a, translocates to the inner nuclear membrane directed by VEGF-C-mediated SUMOylation of lysine-892 on the Nrp2b C-terminus [115]. Within the nucleus, Nrp2b complexes with AR and nuclear pore proteins to directly (KLF4) and indirectly regulate AR-mediated gene expression [116].

Angiogenesis is an essential component for the survival of tumors and metastases. Through interactions with VEGF-A and VEGF-C, Nrp2 contributes to the aggressive nature of cancer cells [117]. In addition, vascular expression of Nrp2 promotes angiogenesis and endothelial cell migration in pancreatic neuroendocrine tumors through a VEGF/VEGFR2-independent pathway by activating the SSH1/cofilin/actin axis [118]. Selective depletion of Nrp2 on endothelial cells (*Nrp1/2*^flfl^*PDGFb*.iCreER) inhibited primary tumor growth, as well as metastasis and secondary site angiogenesis by inducing the rapid transport and degradation of VEGFR2 to Rab7^+^ endosomes [119]. By contrast, Sema3-Nrp2 signaling may inhibit cancer cell migration, repressing tumor growth and metastasis [7]. Sema3F is a potent inhibitor of tumor angiogenesis and metastasis where it inactivates RhoA, depolymerizes F-actin, and suppresses tumor cell migration. Under hypoxic conditions, cancer cells downregulate transcription of Nrp2, preventing antitumor effects of Sema3F [120]. In conjunction, loss of Nrp2 by cancer cells increased VEGF levels in culture, resulting in VEGFR2 phosphorylation and activation of downstream signaling by MAPKs to promote endothelial cell migration and sprouting [120].

The interaction of Nrp2 with integrins on cancer cells and endothelial cells promotes both tumor progression and metastasis. Within the tumor microenvironment, Nrp2 in endothelial cells promotes lymphangiogenesis by activating the integrin-α9β1/FAK/Erk pathway-independent VEGF-C/VEGFR3 signaling [113]. Nrp2 is preferentially expressed on cancer stem cells capable of initiating tumors where it activates α6β1 integrin expression [39]. This results in an autocrine feedback loop whereby α6β1 integrins drive FAK-mediated activation of Ras/MEK and subsequent expression of the Hedgehog effector GLI1, which upregulates Nrp2 [39,114]. GLI1, expressed by 76% of lung cancers, can be noncanonically activated by Nrp2/VEGF-mediated MAPK/ERK signaling, and silencing GLI1 attenuates cancer cell stemness and proliferation, and increases their susceptibility to apoptosis [115]. Cancer cells with high levels of both Nrp2 and α6β1 integrin form more focal adhesions with laminin, and that adhesive strength on laminin is Nrp2-dependent [121]. Nrp2 is required for the interaction of α6β1 integrin with the cytoskeleton and cell motility. Collectively, the Nrp2/α6β1 integrin axis drives tumor initiation and stem cell-like properties in cancer cells. Nrp2 on cancer cells also interacts with α5 integrins on endothelial cells to mediate vascular extravasation and metastasis, and in renal cell carcinoma, is positively correlated with tumor grade and is highest in metastatic tumors [95].

### 6.4. Tumor-Associated Macrophages and Neuropilin-2

Little is known about the role of Nrp2 in TAMs during lung cancer. Nrp2 is upregulated in pMacs adjacent to lung cancer margins, with reduced expression in pMacs in distant normal lung tissue [45]. As TAMs are highly immunosuppressive, this suggests that Nrp2 may be involved in the ongoing immune escape of lung cancer. Nevertheless, insight may be gained from examining other cancers. Nrp2 is a positive regulator of Hedgehog signal transduction, promoting signaling in a temporal and cell-specific spatial basis [122,123]. In response to cancer-cell-secreted Sonic Hedgehog (SHH), Nrp2-mediated signaling promotes an immunosuppressive (alternatively activated) polarization of TAMs through the transcription factor Krüppel-like factor 4 (Klf4) [124]. Depletion of Nrp2 in TAMs (*Nrp2*^fl/fl^*CSF1R*-iCre) inhibited efferocytosis and transcription of anti-inflammatory factors IL-10, IL-4, TGFβ, PD-L2 while promoting IL-12 and IFNβ expression, collectively slowing tumor growth in mouse pancreatic cancer [46,86]. Our group has recently shown that Nrp2 is positively correlated with macrophage infiltration within paired primary and metastatic tumors from patients with breast cancer [46]. Within malignant pleural effusions (MPEs), pleural fluid containing cancer and immune cells secondary to metastasis, macrophages which expressed high levels of Nrp2 had significantly increased Tie2 (pro-angiogenic), IRF5 and CD86 (pro-inflammatory), PD-L1 (immunosuppressive), and CX_3_CR1 (tissue homing) [46]. Notably, Nrp2 high macrophages lacked Nrp1 expression, suggesting that Nrp2 and Nrp1 are associated with discrete macrophage populations in breast cancer MPEs [46]. Genetic depletion of either Nrp2a or Nrp2b in macrophages in vitro inhibited phagocytosis/endosomal processing of apoptotic tumor cells, increased tumor cell migration in response to culture with Nrp2a^KO^/Nrp2b^KO^ macrophage supernatants, and decreased IL-10 production in response to LPS [46]. Notably, loss of either Nrp2 isoform did not affect pro-inflammatory TNFα, IL-12, or chemoattractant CCL2.

## 7. Cardiac Disease

### 7.1. Neuropilin-2 in Angiogenesis and Lymphangiogenesis

Within the cardiovascular system, Nrp2 is present in the smooth muscle of blood vessels, cardiac muscles, and nodes of the heart which control blood pressure and heart rate. Here, Nrp2 regulates endothelial cell adhesion, migration, and permeability, directing angiogenesis during both homeostasis and disease [28,29]. Nrp2/VEGF signaling is a potent driver of angiogenesis. Expressed during hematopoietic differentiation, the transcription factors GATA-binding protein 2 (GATA2) and LIM domain only 2 (Lmo2) have been identified in endothelial cells and are upregulated upon VEGF-induced angiogenesis [125]. Inhibition of GATA2/Lmo2 in primary endothelial cells inhibits VEGF-induced migration and sprouting due to loss of Nrp2 [125]. Overexpression of Nrp2 in GATA2/Lmo2-deficient endothelial cells restored the response to VEGF, demonstrating that GATA2/Lmo2 regulate VEGF-induced angiogenesis via Nrp2-dependent mechanisms [126]. In addition to signaling independently of VEGFR2, Nrp2 promotes VEGF-A/VEGFR2 interactions, leading to VEGFR2 phosphorylation and enhanced cell survival and migratory signaling cascades [119].

Nrp2 can promote angiogenesis independent of VEGFs. Nrp2 regulates actin polymerization, controlled in part by cofilins, which are actin-binding proteins that regulate actin filament dynamics and the reorganization of actin structures. Loss of cofilin inhibits human umbilical vein endothelial cell (HUVEC) migration and angiogenesis [127]. Nrp2 activates cofilin to induce F-acting polymerization via cofilin phosphatase slingshot-1 (SSH1), and silencing SSH1 ameliorates Nrp2-directed HUVEC migration and F-actin polymerization [118]. Unlike Nrp1, which interacts with b3-integrins and VEGF to promote cell motility, Nrp2 induces cytoskeletal remodeling through interactions with a5-integrin and its extracellular matrix (ECM) ligand fibronectin [28]. Here, Nrp2 activates Rab11, a master regulator of intracellular membrane trafficking, to induce Rac1-mediated recycling of a5-integrin–phospho-FAK complexes to new adhesion sites, leading to endothelial cell adhesion and migration [28,128]. It has also been shown that Nrp2 depletion and associated reduction in focal adhesion turnover leads to increased growth of tensin-1 positive fibrillar adhesion and subsequent fibronectin fibrillogenesis. In a model of conditional knockdown of Nrp2 in endothelial cells (*Nrp2*^fl/fl^*PDGFb*-iCre), Nrp2 was shown to be essential for physiological vascularization of the post-natal retina [119,128].

Endothelial cell migration is essential for angiogenesis and is mediated by chemotactic (e.g., VEGF), haptotactic (substrate-bound signals, e.g., ECM), and mechanotactic (e.g., shear stress, substrate stiffness) stimuli. Endothelial cell migration further requires the degradation of ECM to facilitate movement [129]. In response to Sema3G, SMCs produce MMP2 which, in a Nrp2/PlexinA1-dependent manner, potentiates proliferation and migration [130]. Here, Sema3G activates YAP via Nrp2/PlexinA1, but inhibits YAP with Verteporfin YAP activation and reduced subsequent cyclin D1/E expression [130]. The ligand-inducible transcription factor PPARγ, involved in lipid and glucose metabolism, has been implicated in endothelial cell migration during vascular development. PPARγ was shown to upregulate Sema3G transcription, which promotes endothelial cell motility through Nrp2 signaling, and inhibition of either Nrp2 or Sema3G abrogates cell migration [131]. Collectively, these studies illustrate the multiple mechanisms by which Nrp2 controls important developmental and functional aspects of blood vessels.

Following tissue injury, neuropilins are essential components for angiogenesis and vascular development. During carotid balloon injury in rats, in which endothelial denudation and arterial stretch induce neointimal hyperplasia involving vascular smooth muscle cell (VSMC) migration and proliferation, Nrp2 was shown to promote re-endothelialization and increase hyperplasia after injury [32]. Both Nrp2 and Nrp1 phosphorylated PDGFα and PDGFβ to stimulate VSMC migration, but only Nrp2 induced VSMC proliferation [32]. In a model of diabetic wounds, the transcription factor Forkhead box M1 (FOXM1), Sema3C, and Nrp2 levels were repressed and associated with delayed wound healing. Conversely, FOXM1 overexpression promoted macrophage recruitment and alternatively activated polarization via the Sema3C/Nrp2/Hedgehog signaling axis to accelerate wound healing [124]. Corneal injury induces the expression of Sema3A, Sema3C, and their receptor Nrp2, and thus, the local administration of siRNA and monoclonal antibodies (mAbs) targeting Sema3C or Nrp2 inhibited wound healing and nerve fiber regeneration [132]. Nrp2 regulates vascular permeability by the binding of angiopoietin-like 4 (ANGPTL4), an HIF-1-regulaged gene product, to activate RhoA/Rock signaling and subsequently breakdown junctions between endothelial cells and increase permeability [133]. Lastly, in response to Sema3F, an endogenous angiogenesis and lymphangiogenesis inhibitor, Nrp2 modulates tissue swelling and the resolution of post-inflammation edema [48].

The lymphatic system mediates the induction, progression, and resolution of tissue inflammation, and provides an anatomical framework for immune surveillance [126]. Nrp2 on endothelial cells regulates the development of lymphatic capillaries and lymphatic valves, primarily through VEGF-C, which are indispensable for normal drainage of interstitial fluid [28]. Lymphatic vessel patterning is regulated by blood vessels during development, and artery-derived Sema3G and endothelial cell receptor PlexinD1 complex with Nrp2 to promote lymphatic branching and development [134]. Mice that are genetically ablated of Nrp2 have significantly fewer or a complete absence of small lymphatic vessels and capillaries in the heart and lung. The small number of lymphatic vessels that develop in Nrp2^KO^ mice are incorrectly positioned and in some cases are abnormally large. Compared to wild-type littermates, vascular permeability was 2.5-fold greater in Nrp2^KO^ mice, which developed massive tissue swelling and edema after delayed-type hypersensitivity reactions [48]. The addition of exogenous Sema3F inhibited vascular permeability, implicating Nrp2 in regulating vessel leakage. In addition, the loss of Nrp2 reduced lymphatic endothelial cell proliferation. In the context of a highly coordinated response, Nrp2 decreases blood vessel permeability, which increases lymphatic vessel drainage to modulate tissue swelling, control the movement of immune cells, and resolve edema following inflammation.

### 7.2. Neuropilin-2 in Cardiovascular Disease

Recent studies have begun to shed light on the role of Nrp2 in cardiovascular disease and intimal hyperplasia, expertly reviewed by Harman et al. [8]. During homeostasis, Nrp2 is expressed at low levels on vascular smooth muscle cells (VSMCs) but is rapidly upregulated in response to arterial injury or inflammation [32,135]. VSMCs are the primary mediators of vascular remodeling and, in a study following acute carotid artery injury in rats, Nrp2 expression was upregulated, resulting in neointimal hyperplasia [32]. Reduction in Nrp2 after injury inhibited neointimal thickening [32]. Smad3, a canonical signaling protein of TGFβ, positively regulates the transcription of Nrp2, and Nrp2/Smad3 signaling induces SMC proliferation, migration, and dedifferentiation [136]. Smad3/Nrp2 signaling inhibition (either via Nrp2 silencing or in Smad3-haploinsufficient mice) demonstrated reduced intimal hyperplasia by 47% in a mouse model of arterial injury [136]. Other studies have implicated Nrp2 in endothelial-to-mesenchymal transition during occlusive vascular disease, following activation of Nrp2/TGFβ/TGFβR1 complexes [8]. α5 and α9 integrins bind numerous proteins, including fibronectin, fibrinogen, and vitronectin, which are enriched in the ECM deposited during vascular remodeling. In addition to promoting vascular adhesion and extravasation, Nrp2/α5β1 integrin complexes with fibronectin are implicated in cardiovascular disease-associated NF-κβ signaling with subsequent increases in pro-inflammatory cytokines, chemokines, and adhesion molecules [95,113]. Therefore, Nrp2 may orchestrate downstream signaling via integrin-mediated ECM or intercellular interaction to adapt to the microenvironment.

Contrary to their conventional function through Plexin/Sema3 complexes, Nrp2 may inhibit neovascularization while promoting vascular permeability. Here, Nrp2/Plexin/Sema3G signaling leads to release from the ECM via inactivation of β1-integrin [134,137]. Sema3F/Nrp2 signaling inhibits PI3κ and Akt activity, represses mTOR-dependent RhoA GTPase, and decreases VEGF and cytoskeleton stability, which inhibit endothelial cell proliferation and survival [138]. Nrp2^KO^ mice experience increased vascular permeability and edema during acute inflammation, which may result from the lack of Sema3F signaling [48].

In regions of vessels with disturbed blood flow, low sheer stress contributes to disordered endothelial cell arrangement and impaired function. Nrp2 has been shown to be upregulated in endothelial cells of mice and HUVECs under low sheer stress, and knockdown of Nrp2 in Apoe-/- mice mitigated the development of atherosclerosis [139]. Furthermore, in blood vessels with low shear stress, Nrp2 activation leads to increased endothelial cell apoptosis and depletes cellular energy through increased PARP1 expression [139]. Mechanistically, GATA2 increases Nrp2 under low sheer stress, which subsequently promotes endothelial cell apoptosis and atherosclerosis via increased PARP1 expression [139]. In a mouse model of oxygen-induced retinopathy, HIF-2α-induced Sema3G/Nrp2/PlexinD1 complexes increased β-catenin and VE-cadherin interactions in the endothelium, leading to reparative vascular remodeling and the amelioration of ischemic retinopathy [140].

Transmigration of monocytes via blood vessels and their subsequent differentiation into macrophages are fundamental to both tissue maintenance and atherosclerosis. Here, Sema3F promotes vascular expression of the cellular adhesion molecule PECAM-1 in a Nrp2-dependent manner in both mouse models of peritonitis and human mononuclear cell diapedesis through endothelium in vitro [141]. Monocyte-derived macrophages are recruited to plaques where they either promote resolution or progression of atherosclerotic lesions. Macrophages within plaques are heterogenous in phenotype and function, imprinted by their various origins, including circulatory monocytes, tissue-resident macrophages of the vasculature, or trans-differentiation of VSMCs [142,143]. Within the atherosclerotic lesion, free fatty acids and oxidized low-density lipoprotein specifically are ingested by macrophages, which are then termed foam cells. Free fatty acids stimulate TLR4 signaling and NF-κβ activation, leading to upregulation of TNFα, IL-1β, IL-6, and COX-2-derived prostaglandin E2 [144,145]. Proinflammatory cytokines induce lipolysis and the further production of free fatty acids, establishing a positive-feedback loop sustaining chronic inflammation [146]. In conjunction with promoting apoptosis within atherosclerotic lesions, foam cells have a reduced capacity to phagocytose and remove dead or damaged cells, leading to secondary release of DAMPs, further inflammation, and destabilization of the plaque [145,147]. Rupture of the fibrous cap containing the plaque may trigger blood clotting with subsequent complications such as heart attack and stroke. In mouse models of atherosclerosis, depletion of Nrp2 reduces plaque size and lipid content [139]. Within the plasma of patients who recently survived out of hospital cardiac arrest, marked by inflammatory pathophysiology, Sema3F was significantly elevated compared to those with stable coronary artery disease or healthy volunteers, and was associated with decreased survival, myocardial dysfunction, and prolonged vasopressor therapy [141].

### 7.3. Neuropilin-2 in Heart Failure

Nrp1 has an important role in maintaining normal cardiac function, with genetic ablation in cardiomyocytes and VSMCs resulting in cardiomyopathy, aggravated ischemia-induced heart failure, and hereditary hemorrhagic arteriovenous malformations. However, much less is known about the role of Nrp2 in cardiac development and function [48]. Nrp2 is expressed on mesodermal precursor cell subpopulations derived from blood islands, primitive streak, and lateral plate mesoderm that differentiate into cardiomyocytes, endothelium, and VSMCs [8,148]. As the mesoderm differentiates into VSMCs, Nrp2 expression is lost and is re-expressed in response to disease or injury [32,149,150]. Nrp2 is highly expressed in the endothelial cells of veins and is essential for VEGFR-2 phosphorylation and endothelial cell survival and migration [29,148,151]. Nrp2 expression by activated endothelial cells and VSMCs may interact with integrins expressed by monocytes to mediate extravasation of monocytes into vessels [28]. Nrp2 deficiency leads to increased vascular permeability with resultant edema [48]. Nrp2 was one of four plasma proteins, along with Beta 2 microglobulin, alpha-1-antichymotrypsin, and complement component C9, that were increased in heart failure patients in relation to disease severity and pulmonary dysfunction [152]. Additionally, plasma Nrp2 levels were predictive of heart failure rehospitalizations in patients with preserved ejection fraction, and all-cause mortality in patients with reduced or preserved ejection fraction [153].

## 8. Therapeutic Targeting of Nrp2

Given the context-dependent functionality of Nrp2, establishing strategies to leverage its expression for the treatment of disease may require targeting of specific cells or tissues, co-modulation with additional mediators, and careful selection of disease/cell state. A brief list of trials targeting Nrp2 is listed in Table 1. Efzofitimod is a first-in-class biologic immunomodulator targeting Nrp2 that is in clinical development for the treatment of interstitial lung disease (ILD), a group of immune-mediated disorders that can cause inflammation and progressive fibrosis, or scarring, of the lungs [154]. It is a tRNA synthetase-derived therapy based on a naturally occurring, lung-enriched, splice variant of histidyl-tRNA synthetase (HARS) [155]. Efzofitimod selectively modulates activated myeloid cells, predominantly macrophages, which upregulate Nrp2 to resolve inflammation and potentially prevent the progression of fibrosis [156,157]. The proposed mechanism-of-action for this Nrp2 agonist is to promote the differentiation of anti-inflammatory macrophages to downregulate multiple pro-inflammatory cytokines, including TNFα, IL-6, and CCL2, and receptors such as CD14, that are dysregulated in ILD. In multiple animal models of ILD, efzofitimod reduced inflammation and fibrosis and prevented sarcoid granuloma formation in vitro [158].

Clinical proof-of-concept was established for efzofitimod in a phase 1b/2a study in patients with pulmonary sarcoidosis (NCT03824392) [156,157]. Pulmonary sarcoidosis is an inflammatory disease characterized by a noncaseating epithelioid cell granuloma enriched in macrophages which is treated with corticosteroids [159]. Using an Nrp2-specific antibody developed for clinical diagnosis, pulmonary granulomas but not the surrounding tissues were found to express Nrp2 [160]. Staining further revealed that CD68 (general) and CD163 (alternatively activated) macrophages within granulomas expressed high levels of Nrp2, potentially to limit ongoing inflammation [160]. Systemic delivery of efzofitimod given every 4 weeks for a duration of 24 weeks was safe and well tolerated in 37 patients with pulmonary sarcoidosis and resulted in a dose-dependent, baseline-adjusted steroid reduction of between 5 and 22%. Notably, efzofitimod treatment led to a clinically meaningful, although non-significant, increase in lung function as well as patient-reported quality of life outcomes. This first in-human trial found that the adverse event profiles between efzofitimod and placebo were similar, which provides encouraging signs towards its safety profile and potential use in long-term therapy. Nevertheless, study findings should be interpreted cautiously due to the small sample size, and results of an ongoing phase 3 trial evaluating the efficacy and safety of a 48-week regimen of efzofitimod will shed light on its therapeutic potential.

Numerous preclinical studies have demonstrated the efficacy of Nrp2 inhibition for the treatment of cancers. Administration of Nrp2-blocking antibodies in vivo inhibited tumor vascularity and growth in a mouse model of pancreatic neuroendocrine tumors, repressing F-acting-mediated motility via decreased SSH1-cofilin signaling [118]. The antibody N2E4 blocks the interaction of Nrp2 with β1-integrins and inhibits FAK/Erk/HIF-1α/VEGF signaling [161]. Systemic N2E4 reduced growth and metastasis of pancreatic ductal adenocarcinoma cell lines implanted subcutaneously in mice [161]. Anti-Nrp2 antibody treatment inhibited the growth of human triple negative breast cancer cells in immunocompromised mice, impeding an autocrine feedback loop of Nrp2/α6β1/GLI1/BMI-1 [39]. Antibody-mediated blocking of Nrp2 inhibits CD44^+^ CD24^Low^ cancer stem cells and reduces expression of *ZEB1*, a master regulator of EMT [162]. Further, treatment with a humanized monoclonal antibody, aNRP2-10 (ATYR2810), sensitized triple negative breast cancer (TNBC) cells to cisplatin and/or 5-fluorouracil-induced cytotoxicity in cell lines, patient-derived 3D organoids, and TNBC tumor growth and metastasis in xenograft mouse models [162]. As VEGF/Nrp2 signaling sustains PD-L1 expression on tumor cells, inhibition of VEGF binding via Nrp2-specific monoclonal antibodies (aNRP2-10/humanized and aNRP2-28/mouse) in neuroendocrine prostate cancer organoids and syngeneic prostate cancer mouse models resulted in decreased PD-L1 expression on tumors and increased immune infiltrate and immune-mediated tumor cell killing, while regressing tumors in vivo [110]. Antibody-mediated disruption of Nrp2/VEGF-C has been found to reduce tumor lymphangiogenesis of murine mammary carcinoma and glioblastoma cell lines and was thought to inhibit metastasis by delaying the departure of tumor cells from the primary tumor [93].

Separately, inhibiting Nrp2 may also have benefits for the treatment of occlusive vascular diseases (via VEGF-C/Nrp2 signaling, PDGFβ, and TGFβ signaling). By contrast, inhibiting Nrp2 in macrophages may prolong pathologic inflammation and restrict phagocytosis with deleterious downstream effects for diseases of the heart and lung. The recent characterization of a defined Nrp2 binding pocket with limited potential for ligand interactions will hopefully facilitate the development of novel Nrp2-targeted small-molecule inhibitors [109]. Collectively, preclinical, and clinical trials suggest that Nrp2 may be manipulated either for silencing or amplification of associated signaling pathways for the treatment of numerous diseases. Nevertheless, therapeutic targeting of Nrp2 will likely be dependent upon specific disease pathobiology including diverse tissues, cells, and microenvironmental contexts.

## 9. Conclusions

Nrp2 plays an essential role in embryonic development, blood vessel development, and lymphatic vessel development. Furthermore, Nrp2 is an important modulator in the immune system, macrophages in particular, and is upregulated during pro-inflammatory states and several diseases (Figure 2). These diseases include multiple pulmonary and cardiovascular diseases, ranging from lung cancer to heart failure with its upregulation in lung cancer conferring a worse outcome. Yet, our understanding of Nrp2 is still in its infancy. Its promiscuous binding properties, together with the ability to recruit various downstream signaling molecules, suggest that Nrp2 acts as a secondary, nonredundant factor in biologic processes. At times, as with VEGF interactions, Nrp2 serves to amplify microenvironmental cues. Alternatively, Nrp2 may initiate divergent functions illustrated by Nrp2’s inhibition of NF-κβ in the presence of inflammation. Leveraging Nrp2 for the treatment of disease may require targeted delivery to specific cells, local administration to tissues, or potentially co-administration with other agents to promote synergistic effects.

## 10. Methods

The NIH National Library of Medicine’s PubMed was searched up to 15 January 2024 using the keyword “neuropilin-2”. A total of 586 articles containing “neuropilin-2” were present and reviewed by the authors for suitability. Articles pertaining to neoropilin-2 in heart disease, lung disease, cancer, immunity, and basic function and structure were included in this review. The NIH ClinicalTrials.gov was queried for trials targeting or involving neuropilin-2 and pertinent trials described. The figures were created with BioRender.com.

## Figures and Tables

**Figure 1 jcm-13-01446-f001:**
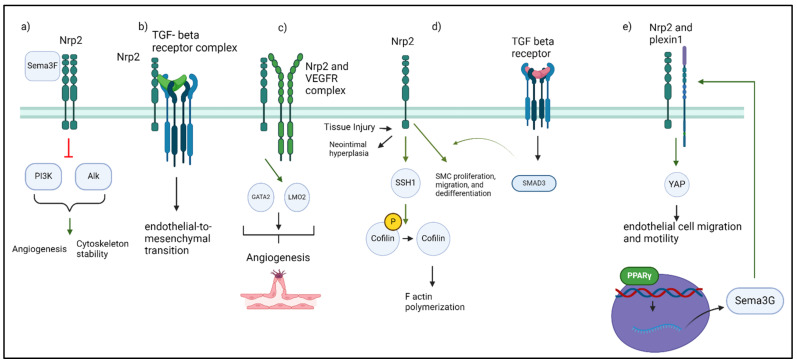
Prominent neuropilin-2 signaling pathways and biologic effects: (**a**) After binding SEMA3F, NRP2 prevents angiogenesis and decreases cytoskeleton stability by inhibiting PI3K and Alk. (**b**) NRP2 and TGF-β receptor complex promotes endothelial-mesenchymal transition. (**c**) NRP2 and VEGFR complex promote angiogenesis through upregulation of GATA2 and LMO2. (**d**) Tissue injury upregulates NRP2 leading to neointimal hyperplasia. NRP2 upregulates SSH1 to dephosphorylate Cofilin, leading to increased F actin polymerization. NRP2 using Smad 3/ TGF-β receptor interaction promotes SMC migration, proliferation, and dedifferentiation. (**e**) PPAR-γ transcriptionally upregulates Sema3G which binds to NRP2/plexin 1 complex to upregulate Yap and promote endothelial cell migration and motility.

**Figure 2 jcm-13-01446-f002:**
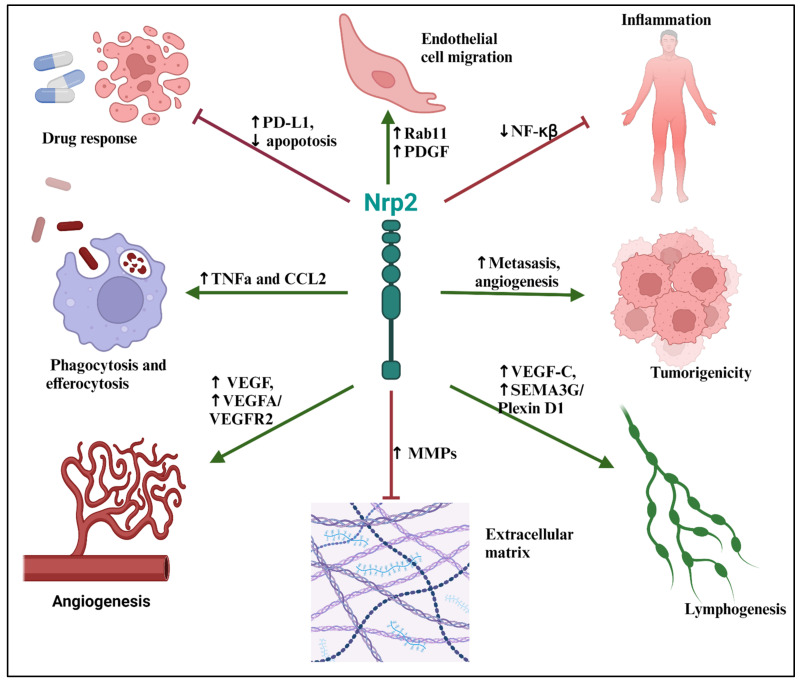
Mechanisms of neuropilin-2-mediated regulation of tissue homeostasis and cardiothoracic disease.

**Table 1 jcm-13-01446-t001:** Current clinical and preclinical trials targeting neuropilin-2.

Disease	Target Pathway	Endpoint	Current Status
Sarcoidosis	Targets macrophages in pulmonary granulomas to potentially downregulate pro-inflammatory cytokines like TNF alpha and IL-6	Weaning of steroid dose and duration Symptomatic quality of life improvement	Currently phase 3 trial taking place using efzofitimod
Systemic sclerosis (SSc)-related interstitial lung disease	Innate immune system Downregulation of inflammatory cytokines	Change and improvement of Pulmonary function tests Symptomatic quality of life improvement	Ongoing phase 2 trial using efzofitimod
Pancreatic neuroendocrine tumor	F-acting-mediated motility via SSH1-cofilin signaling	Tumor vascularity and growth	Preclinical studies
Pancreatic adenocarcinoma	FAK/Erk/HIF-1α/VEGF signaling	Tumor growth and metastasis	Preclinical studies
Prostate cancer	PD-L1 expression due to VEGF/Nrp2 signaling	Decreased PD-L1 expression, increase in immune mediated tumor killing	Preclinical studies
Breast cancer	Nrp2/VEGF-C pathway	Decreased tumor lymphangiogenesis	Preclinical studies
Vascular occlusive disease	VEGF-C/NRP2 pathway	Decrease in vascular inflammation	Preclinical studies

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
