# Peer review of "Orchestrating Resilience: How Neuropilin-2 and Macrophages Contribute to Cardiothoracic Disease"

_jcm, 2024, doi:10.3390/jcm13051446_

Round 1

Reviewer 1 Report

Comments and Suggestions for Authors

The article highlights the use of efzofitimod, a first-in-class biologic immunomodulator targeting Nrp2, in the treatment of interstitial lung disease (ILD). The Nrp2 agonist efzofitimod is described as selectively modulating activated myeloid cells, predominantly macrophages, to resolve inflammation and potentially prevent fibrosis. The present study presents some limitations and Authors are asked to answer some questions:

However, a critical examination prompts consideration of several aspects. Firstly, while the Phase 1b/2a study in patients with pulmonary sarcoidosis (NCT03824392) provides encouraging results, the article could delve deeper into potential limitations and challenges faced during the clinical trial, such as sample size, duration, and long-term safety.

Secondly, the efficacy of efzofitimod in promoting clinically meaningful, though non-significant, increases in lung function and quality of life outcomes raises questions about the clinical significance of these changes. The article would benefit from a discussion on the practical implications of these improvements for patients and whether they translate into meaningful benefits.

Considering the multifaceted functionality of Nrp2 and its potential role in both promoting anti-inflammatory responses and inhibiting cancer growth, how can therapeutic strategies be tailored to achieve a balance that addresses specific disease pathobiology without inadvertently exacerbating detrimental effects in certain tissues or cells? Additionally, are there ongoing investigations or emerging insights into the long-term safety and efficacy of efzofitimod and other Nrp2-targeted therapies that could shed light on their viability as long-term treatment options?

Author Response

We appreciate the Reviewer's positive assessment of our manuscript and have tried to address all concerns fully. To that end, we have: 

  • Described the limitations of the NCT03824392 study including small sample size, limited study duration, and lack of statistical significance in study endpoints beyond safety.
  • We agree that the preliminary data from NCT03824392 must be interpreted cautiously and suggest that upcoming results of the ongoing Phase 3 trial with ATYR1923 will shed further light on treatment efficacy. 
  • The final question about how to leverage Nrp2 for treatment is less straight forward. Whereas, the ongoing Phase 3 trial with ATYR1923 (48 months) will provide data on the safety of systemic Nrp2 inhibition, truly targeted applications of Nrp2 agonists or antagonists are more difficult to envision clinically. We attempt to address this issue in the newly added conclusions statement suggesting that eventual clinical application of Nrp2 targeted therapies will need to be specific to cell type (nanoparticle targeting, etc.), localized tissue treatments (intratumoral, respiratory) or provided in combination with other agents to potentiate Nrp2 inhibition such as T cell checkpoint blockade or targeted radiation. 

Thank you for your time and thoughtful comments which have helped us improved our work. 

Reviewer 2 Report

Comments and Suggestions for Authors

The article is generally well-written and addresses a very interesting and clinically applicable topic. However, several major issues need to be addressed:

1. The methodology of conducting the review is not provided. It seems to be a simple narrative review, but readers should be informed about how the sources were selected, which databases were reviewed, and which keywords or phrases were used. It would be beneficial to follow the typical structure of scientific texts, including an introduction followed by a methods section containing these elements.

2. The review is presented as a monolithic block of text, making it difficult to follow and cumbersome in some areas.

   - Adding attractive figures to illustrate described mechanisms and to summarize pleiotropic effects of neuropilin-2 in cardiac disease would enhance comprehension.

   - Additionally, a well-structured table summarizing therapeutic targeting would be beneficial.

3. A brief summary is lacking.

4. There is no declaration of conflicts of interest, which should be included in reviews as well.

Comments on the Quality of English Language

Typos are present in the text and should be carefully reviewed. For example, in subtitle 6.4, "nueropilin" should be corrected to "neuropilin."

Author Response

We thank the Reviewer for their thoughtful and constructive comments and have attempted to address them all fully. Specific points are listed below. 

  • We have included a methodology section and attempted to better outline the text.
  • We agree the text only was a bit dull. We have added two new figures and one table to summarize the effects of Nrp2 on cardiothoracic disease. 
  • We have included a brief summary (conclusions) section. 
  • Thank you for noting our omission. We have included a conflict of interest statement for all authors. 
  • We have carefully vetted the manuscript for spelling and grammatical errors. Thank you. 

Thank you very much for your time and constructive critique. We hope we have addressed your concerns and believe we have improved this work accordingly.